# Antibiotic Prescriptions among China Ambulatory Care Visits of Pregnant Women: A Nationwide Cross-Sectional Study

**DOI:** 10.3390/antibiotics10050601

**Published:** 2021-05-19

**Authors:** Houyu Zhao, Mei Zhang, Jiaming Bian, Siyan Zhan

**Affiliations:** 1Department of Epidemiology and Biostatistics, School of Public Health, Peking University, Beijing 100191, China; houyu-zhao@pku.edu.cn; 2Department of Pharmacology, The 7th Medical Center of Chinese PLA General Hospital, Beijing 100700, China; may5353@163.com; 3Research Center of Clinical Epidemiology, Peking University Third Hospital, Beijing 100191, China; 4Center for Intelligent Public Health, Institute for Artificial Intelligence, Peking University, Beijing 100871, China

**Keywords:** antibiotics, pregnancy, outpatients

## Abstract

Background: Antibiotic use in pregnant women at the national level has rarely been reported in China. Objectives: We aimed to investigate antibiotic prescriptions during pregnancy in ambulatory care settings in China. Methods: Data of 4,574,961 ambulatory care visits of pregnant women from October 2014 to April 2018 were analyzed. Percentages of Antibiotic prescriptions by different subgroups and various diagnosis categories and proportions of inappropriate antibiotic prescriptions for different subgroups were estimated. Food and Drug Administration (FDA) pregnancy categories were used to describe the antibiotic prescription patterns. The 95% confidence intervals (CIs) were estimated using the Clopper––Pearson method or Goodman method. Results: Among the 4,574,961 outpatient visits during pregnancy, 2.0% (92,514 visits; 95% CI, 2.0–2.0%) were prescribed at least one antibiotic. The percentage of antibiotic prescriptions for pregnant women aged >40 years was 4.9% (95% CI, 4.7–5.0%), whereas that for pregnant women aged 26–30 years was 1.5% (95% CI, 1.4–1.5%). In addition, percentages of antibiotic prescriptions varied among different trimesters of pregnancy, which were 5.4% (95% CI, 5.3–5.4%) for the visits in the first trimester of pregnancy and 0.5% (95% CI, 0.4–0.5%) in the third trimester of pregnancy. Furthermore, the percentages of antibiotic prescriptions substantially varied among different diagnosis categories and nearly three-quarters of antibiotic prescriptions had no clear indications and thus might be inappropriate. In total, 130,308 individual antibiotics were prescribed; among these, 60.4% (95% CI, 60.0–60.8%) belonged to FDA category B, 2.7% (95% CI, 2.1–3.5%) were classified as FDA category D and 16.8% (95% CI, 16.2–17.4%) were not assigned any FDA pregnancy category. Conclusions: Antibiotic prescriptions in ambulatory care during pregnancy were not highly prevalent in mainland China. However, a substantial proportion of antibiotics might have been prescribed without adequate indications. Antibiotics whose fetal safety has not been sufficiently illustrated were widely used in pregnant women.

## 1. Introduction

Antibiotics are among the most commonly prescribed medications in pregnant women for infectious diseases [1,2], which, if untreated, would increase maternal and neonatal mortality and significant fetal risks [1,2,3]. In some developed countries, the proportion of pregnant women receiving at least one antibiotic prescription during pregnancy can be as high as 50% [4,5]. However, antibiotic use during pregnancy may also increase the risk of adverse pregnancy outcomes including spontaneous abortion [6] and diseases in early childhood, such as asthma and epilepsy [7]. Since the thalidomide tragedy, drug use during pregnancy has become a concern for both pregnant women and physicians [8]. In addition, excessive and inappropriate antibiotic use can substantially accelerate antimicrobial resistance (AMR) [6,9], which is recognized as a global priority health issue [9,10]. For pregnant women, AMR can be particularly concerning because the resistant bacteria may be transmitted to the newborn, causing severe infections [6]. Therefore, it is important that antibiotics are appropriately used during pregnancy, considering both potential risks of untreated infections and harmful effects on fetus [1,11]. As a response to the concern of embryo–fetal risk of drugs used during pregnancy, numerous classification systems have been adopted in different countries for labelling drug safety. The most commonly used classification system is the United States Food and Drug Administration (FDA) letter categorization system [12], which classifies drugs into five categories—A, B, C, D and X—according to potential fetal risks, based on the findings of human and animal studies [1,12].

The appropriate use of antibiotics is a key strategy to ensure maternal health, reduce potential fetal risk and tackle antibiotic resistance [1,6]. Studies on the use and appropriateness of antibiotics during pregnancy are important to inform actions for reducing maternal and neonatal morbidity and mortality and combating antimicrobial resistance. To date, most of the studies on antibiotic use during pregnancy have been conducted in some developed countries. Evidence of the use and appropriateness of antibiotics during pregnancy is particularly scarce at the national level in China. In this milieu, we used data from a nationwide prescription database to investigate the current situation of outpatient antibiotic use among pregnant women in China. Furthermore, a well-established classification approach [13,14,15,16] was used to estimate the percentages of antibiotic prescriptions for common co-occurring diagnosis categories and to assess the appropriateness of outpatient antibiotic prescriptions during pregnancy.

## 2. Materials and Methods

### 2.1. Data Source and Participants

The national database used in this study has been described in our previous studies [15,16]. In brief, clinical records, including outpatient and inpatient prescriptions from 194 hospitals in 128 cities of 31 provincial-level regions have been collected in the database since October 2014. Outpatient diagnosis in the database was in the Chinese narrative free text format [15,16]. Several diagnoses could be written together and separated by punctuation marks [15]. Chemical drugs, including antibiotics, were extracted from the billing system and coded in accordance with the Anatomical Therapeutic Chemical (ATC) classification system. All prescribed antibiotics were eventually dispensed to patients. We linked the information of diagnosis and prescription drugs through a unique identifier, which consisted of hospital code, patient identification number and visit date. All visits to outpatient clinics and emergency departments from October 2014 to April 2018 were included.

Adult pregnant women aged 18–50 years were included in this study and were identified by detecting information on pregnancy in the diagnosis text. We excluded patients with an abnormal pregnancy and those who terminated their pregnancy, for example, induced abortion and ectopic pregnancy. Because for these patients, there was no need to consider the embryo–fetal risk of antibiotics. Pregnancy was divided into three trimesters according to the diagnosis description or gestational week—the first trimester lasted from conception to week 13 of gestation, the second was weeks 14–26 of gestation and the third was from week 27 to the delivery [4].

Fifty-five hospitals that did not submit diagnosis records were excluded from this study [16]. Another 11 hospitals were excluded due to insufficient outpatient visits following the deletion of records with missing data. Comparisons between the included and excluded hospitals are given in the Appendix A. Finally, 128 hospitals (38 secondary and 90 tertiary hospitals) from 28 provinces and 90 cities were included in the final analyses.

### 2.2. Definition of Antibiotics and Outpatient Visits

Antibiotics for systemic use according to ATC code J01 were included in this study. To assess whether potential teratogenic risk and fetal harm were considered when prescribing antibiotics for pregnant women, we classified the antibiotics into categories B, C and D by following the FDA letter categorization system [1,12,17,18]. Antibiotics that had not been assigned to any FDA category were classified as category N in this study [12]. All antibiotics prescribed for pregnant women and the categories in this study are presented in the Appendix A.

Antibacterial spectrum was classified according to previous studies [19,20]. Broad-spectrum antibacterial agents included second- to fourth-generation cephalosporins (ATC J01DC, J01DD and J01DE), fluoroquinolones (J01MA), macrolides (J01FA, except for erythromycin J01FA01), combinations of penicillins (J01CR) and streptomycin (J01GA). Other antibiotics, such as beta-lactamase-sensitive penicillins (J01CE) and first-generation cephalosporins (J01DB), were classified as narrow-spectrum.

Multiple prescriptions of drugs and diagnoses from the same patient on the same day in the same hospital were considered as one visit in this study [16]. Patients with missing diagnoses and demographic information were excluded.

### 2.3. Diagnosis Classification

Following the approach employed in previous studies [13,14,15,16], we classified the first five diagnoses, including pregnancy diagnosis, linked to the same visit into different categories and three tiers: (1) ‘tier 1′ if the condition almost always justifies antibiotic use, such as pneumonia; (2) ‘tier 2′ if the condition only sometimes justifies antibiotic use, such as acute sinusitis; and (3) ‘tier 3′ if the condition almost never justifies antibiotic use, such as sole pregnancy without any co-occurring diagnosis of infectious diseases, through a regular expression-based classification algorithm [15]. Finally, a single diagnosis category was assigned to each prescription after all the first five diagnoses were classified as tier 1, 2, or 3. For multiple diagnoses in the same visit, priority was given to tier 1 diagnosis, followed by tier 2 diagnosis and, finally, tier 3 diagnosis. Details of the diagnosis classification algorithm and diagnosis categories have been described in our previous studies [15,16].

### 2.4. Predictors of Antibiotic Prescribing

We reported percentages of antibiotic prescriptions overall and by different patient or hospital characteristics including age groups (18–25, 26–30, 31–35, 36–40 and 40–50 years), trimester of pregnancy, department of visit (outpatient or emergency department), diagnosis tier (tier 1–3), year (2014–2018) and season of outpatient visits and China’s economic region. According to the National Bureau of Statistics of China [21], mainland China was divided into four economic regions: the Eastern (including Beijing, Tianjin, Hebei, Shandong, Jiangsu, Shanghai, Zhejiang, Fujian, Guangdong and Hainan), Central (including Shanxi, Henan, Anhui, Hunan, Hubei and Jiangxi), Western (including Nei Mongol, Shaanxi, Ningxia, Gansu, Qinghai, Xinjiang, Xizang, Sichuan, Yunnan, Chongqing, Guizhou and Guangxi) and Northeastern (including Heilongjiang, Jilin and Liaoning). Details of the population and health facilities in each region have been described in our previous study [16]. These factors were further analyzed to identify the predictors of antibiotic prescribing in pregnancy using a multi-level multivariate binary logistic model.

### 2.5. Statistical Analysis

The percentages of antibiotic prescriptions were calculated as the percentages of outpatient visits with antibiotic prescriptions for different diagnosis categories and subgroups. Clopper–Pearson exact method was used to calculate the 95% confidence intervals (CIs) of percentages of antibiotic prescriptions for the entire study population, as well as different subgroups. Predictors for antibiotic prescriptions were assessed using a multi-level binary logistic regression, with random intercept for each hospital. Odds ratios with 95% CIs were reported. The proportion of prescribed antibiotics in each FDA pregnancy category among different diagnosis categories and subgroups was estimated according to the prescription frequency. The simultaneous 95% CIs for multinomial proportions were estimated using Goodman method [22]. We also identified the most commonly used individual antibiotics within each FDA pregnancy category, by calculating the proportion of each prescribed antibiotic relative to the total prescribed antibiotics.

Oracle 11gR2 and PLSQL developer 11.0 (Oracle Corp., Redwood Shores, CA, USA) were used for data extraction and diagnosis classification. Statistical analyses were performed using SAS 9.4 (SAS Institute Inc., Cary, NC, USA).

## 3. Results

### 3.1. Basic Characteristics

After excluding the visits of women with the diagnoses of abnormal pregnancy and pregnancy termination, 4,574,961 ambulatory care visits of pregnant women from 90 cities and 28 provinces were included in this study. Among these visits, 2,080,643 (45.5%) involved women aged 26–30 years; 4,502,627 (98.4%) occurred in outpatient clinics, 4,279,852 (93.5%) presented at tertiary-level hospitals. Furthermore, 2,317,388 (50.7%) visits were in the Eastern region and 1,202,141 (26.3%) visits were during winter (Table 1). In addition, most visits (4,430,354, 96.8%) were assigned to a tier 3 diagnosis category, including a single pregnancy description without any co-occurring diagnosis.

### 3.2. Percentages and Predictors of Antibiotic Prescriptions for Pregnant Women

Of all ambulatory care visits of pregnant women, at least one antibiotic was prescribed during 92,514 visits (2.0%; 95% CI, 2.0–2.0%) (Table 2). The percentage of antibiotic prescriptions in outpatient visits was the highest during the first trimester (5.4%; 95% CI, 5.3–5.4%) and the lowest during the third trimester (OR 0.22; 95% CI, 0.21–0.23). Pregnant women aged over 40 years had the highest percentage of antibiotic prescriptions, with 4.9% (3786 visits; 95% CI, 4.7–5.0%) of outpatient visits ended with antibiotics. In contrast, antibiotics were prescribed only during 1.5% (95% CI, 1.4–1.5%) of visits by women aged 26–30 years and 1.8% (95% CI, 1.7–1.8%) of visits by women aged 31–35 years. These values were significantly lower than that for women older than 40 years (adjusted OR 0.25 [95% CI, 0.24–0.26] and 0.33 [95% CI, 0.32–0.35], respectively). Among all visits, antibiotics were prescribed in 5.2% (95% CI, 5.0–5.3%) of the visits in the emergency department, significantly higher than that in outpatient clinics (OR 8.68; 95% CI, 8.30–9.07). Furthermore, antibiotics were prescribed during 1.8% (95% CI, 1.8–1.8%) of tertiary hospital visits and it was lower than that in secondary hospitals (OR 0.38; 95% CI, 0.23–0.61). The percentage of antibiotic prescriptions was the lowest for visits in the Eastern region of China (1.7%; 95% CI, 1.7–1.7%) and in summer (1.8%; 95% CI, 1.8–1.8%). In addition, the results showed a slight temporal trend of antibiotic prescriptions, with the highest prescription percentage in 2014 (3.3%; 95% CI, 3.3–3.4%). The co-occurrence of tier 1 diagnoses was a strong predictor for antibiotic prescriptions and antibiotics were prescribed during 40.4% (95% CI, 39.9–40.9%) of visits for tier 1 diagnoses, which was 31.98 (95% CI, 30.95–33.05) times higher than that during visits for tier 3 diagnoses.

### 3.3. Antibiotic Prescriptions for Various Diagnosis Categories and the Appropriateness of Antibiotic Use

Table 3 shows the percentages of antibiotic prescriptions for various diagnosis categories. Among the 35,442 visits that were assigned to tier 1 diagnoses, antibiotics were prescribed during 57.8% (95% CI, 52.0–63.5%) of the visits for pneumonia and 31.7% (95% CI, 30.4–33.0%) for urinary tract infections. Acute sinusitis and acute pharyngitis were the top two diagnoses with the highest percentages of antibiotic prescriptions during visits of tier 2 diagnoses. Antibiotics were prescribed during 37.8% (95% CI, 31.9–44.0%) of the visits for acute sinusitis and 34.3% (95% CI, 32.3–36.3%) for acute pharyngitis during pregnancy. The overall percentage of antibiotic prescriptions was only 1.6% (95% CI, 1.6–1.6%) for visits of tier 3 diagnoses during pregnancy. However, antibiotic prescription was also prevalent for some tier 3 diagnoses. For example, the percentages of antibiotic prescriptions for patients with non-suppurative otitis media, upper respiratory tract infections and acute bronchitis were 23.1% (95% CI, 14.9–33.1%), 31.3% (95% CI, 30.3–32.2%) and 44.5% (95% CI, 42.4–46.6%), respectively. A quite high percentage of antibiotics were even prescribed for some symptoms or signs, with 28.8% (95%CI, 26.4–31.3%) of visits for fever and 14.4% (95%CI, 12.5–16.5%) for cough in pregnancy resulting in antibiotic prescriptions. Furthermore, for almost all categories of diagnoses, the percentages of antibiotic prescriptions were obviously lower in pregnant women than in general patients of whom 10.9% of visits were associated with antibiotic prescriptions. Among all 92,514 visits with antibiotic prescriptions, 69,611 (75.2%; 95% CI, 74.9–75.6%) were classified as inappropriate, 8590 (9.3%; 95% CI, 8.6–10.1%) were potentially appropriate and 14,313 (15.5%; 95% CI, 14.8–16.2%) were classified as appropriate. Appendix A shows the percentages of antibiotic prescriptions by diagnosis and age group. For pregnant women over 35 years, 40.9% (95% CI, 39.4–42.4%) visits for tier 1 diagnoses, 10.1% (95% CI, 9.6–10.7%) visits for tier 2 diagnoses and 2.5% (95% CI, 2.4–2.5%) visits for tier 3 diagnoses were prescribed antibiotics, all of which were obviously higher than that of pregnant women aged 26–35 years.

### 3.4. Antibiotic Use by the FDA Categories and Antibiotic Prescription Pattern

Of the 92,514 visits with antibiotic prescriptions, a total of 130,308 individual antibiotics were prescribed, of which 78,272 (60.1%; 95% CI, 59.8–60.3%) were broad-spectrum antibiotics (Appendix A). The three most commonly prescribed antibiotics were ornidazole (21,642, 16.6%), levofloxacin (13,068, 10.0%) and azithromycin (11,055, 8.5%) (Appendix A). Among all antibiotics prescribed to pregnant women, 60.4% (95% CI, 60.0–60.8%) belonged to FDA category B, 20.0% (95% CI, 19.5–20.6%) were FDA category C, only 2.7% (95% CI, 2.1–3.5%) were FDA category D and the remaining 16.8% (95% CI, 16.2–17.4%) were not assigned a FDA pregnancy category (Figure 1). Regarding the use of category D antibiotics with evidence of fetal risk in humans, there were obvious heterogeneities in some subgroups. For example, among visits during the first trimester, 3.3% (95% CI, 2.6–4.1%) of the prescribed antibiotics were FDA category D, whereas for visits during the second and third trimesters, the proportion of FDA category D antibiotics was only 0.2%. In addition, the proportion of FDA category D antibiotics was relatively higher for visits in secondary hospitals (4.4%; 95% CI, 3.3–6.0%), in the western region of China (4.1%; 95% CI, 3.2–5.4%) and those involving women aged over 40 years (3.5%; 95% CI, 1.4–8.3%). Furthermore, in the central region, the proportions of both FDA category B and D antibiotics were obviously lower than that in other regions, whereas the prescriptions of antibiotics of category C and N were considerably more prevalent in this region. For different diagnosis categories, 2.6% (95% CI, 1.4–4.7%) of antibiotics for tier 1 diagnoses, 0.9% (95% CI, 0.1–7.2%) for tier 2 diagnoses and 2.9% (95% CI, 2.3–3.8%) for tier 3 diagnoses were FDA category D (Figure 2). Except for a few categories of diagnoses, such as acne, the proportions of FDA category B antibiotics were over 60% for nearly all conditions. Appendix A presents the proportion of each individual antibiotic within different FDA categories and the most commonly prescribed category B antibiotics were azithromycin (14.0% of all category B antibiotics), cefixime (13.0%), cefdinir (12.9%) and metronidazole (11.7%). The most commonly prescribed category C antibiotics were levofloxacin (50.0% of all category C antibiotics) and tinidazole (29.8%). The most commonly prescribed category D antibiotic was etimicin (81.9% of all category D antibiotics). Finally, the most commonly prescribed category N antibiotic was ornidazole (98.7% of all category N antibiotics).

## 4. Discussion

### 4.1. Main Findings of this Study

To the best of our knowledge, this study is the first in China to investigate antibiotic prescriptions for ambulatory care visits of pregnant women using such a big prescription data covering almost all provinces of mainland China. We found that antibiotics were prescribed during 2.0% of ambulatory care visits in Chinese hospitals for pregnant women. However, nearly three-quarters of the antibiotics might have been prescribed without appropriate indications. Age, pregnancy trimester, department and season of visit should be considered as potential determinants of antibiotic prescriptions. The prescribing of FDA category D antibiotics, which are potentially harmful to fetus, was low in China; nevertheless, the prescriptions of antibiotics with unclear fetal risks were relatively more prevalent.

### 4.2. Comparison of Our Findings with Existing Literature

Evidence of antibiotic prescribing for pregnant women at the national level is scarce in China. However, our results that only 2% of ambulatory care visits of pregnant women resulted in antibiotic prescriptions seems obviously lower than the prescription rates in other studies. A survey conducted by Chen et al., in a single hospital indicated that 22.3% of women who delivered in the hospital were prescribed antibiotics during pregnancy [23]. In Taiwan, a study using data from an insurance database reported that 33.7% of pregnant women received antibiotic prescriptions during pregnancy [24]. In developed countries, the prescribing of antibiotics during pregnancy is prevalent and varies considerably. In Germany, 14.7% of pregnant women are prescribed antibiotics [25]. In France and Italy, this percentage is approximately 50% [4,5]. However, in the UK, Canada and the USA, nearly one-third of women receive antibiotics during pregnancy [26,27,28]. Nevertheless, in the Netherlands and some regions of Italy, less than 25% of women are prescribed antibiotics during pregnancy [3,11]. This discrepancy in the prescription rates can be partially attributed to different methods, data sources and populations. For example, most studies in developed countries were based on a cohort design [3,4,11,26,27,28]. Meanwhile, the study in mainland China was based on medication information obtained through a questionnaire survey and participants’ recollection [23] and, thus, the percentages of antibiotic prescriptions were estimated according to the antibiotic use throughout pregnancy. In this study, prescription data were extracted from cross-sectional medical records; thus, some pregnant women might have visited the hospital just for prenatal care and required no medications, resulting in a potential underestimation of the percentage of antibiotic prescriptions. In addition, a recent study using a random sample of insurance data illustrated that the use of overall medications in mainland China was considerably lower than that reported in developed countries [29]. This may be an important reason for low antibiotic prescriptions during outpatient pregnancy visits in mainland China.

We found that for almost all conditions and diagnosis categories, percentages of outpatient antibiotic prescriptions were considerably lower in pregnant women than in general patients in China. For example, 10.9% of visits with all conditions, 30.6% of visits with tier 2 diagnoses and 7.6% of visits with tier 3 diagnoses were prescribed antibiotics among general outpatients [16]; however, the corresponding percentages were only 2.0%, 7.9% and 1.6% in pregnant women. Furthermore, broad-spectrum agents accounted for a lower proportion of antibiotics in pregnant women than in general outpatients [16]. However, the lower percentages of antibiotic prescriptions did not guarantee a higher proportion of appropriate antibiotic prescriptions, as nearly half of antibiotic prescriptions for general patients [16] and three-quarters of antibiotic prescriptions for pregnant women were classified as inappropriate. This finding indicates that physicians might be considerably more cautious when prescribing antibiotics for pregnant women and this cautiousness might have a greater effect on antibiotic prescriptions for conditions that only sometimes justify antibiotic use, resulting in potential undertreatment of some infectious conditions. This undertreatment could have harmful effects on maternal and fetal health [1,2,17], as infections may increase both maternal and neonatal mortality [2] and are associated with an increased risk of adverse pregnancy outcomes including spontaneous abortion, prematurity and low birth weight [1,2,3,17].

Our results suggested that older pregnant women tended to have a higher percentage of antibiotic prescriptions, except pregnant women aged <26 years. Furthermore, pregnant women in their first trimester had the highest risk of receiving antibiotic prescriptions. These results were in agreement with those of previous studies, indicating that pregnant women used more drugs, including antibiotics, in the first trimester [24,28,29]. However, several studies have reported that antibiotic use is lower in the first trimester [30,31]. In addition, a few studies have indicated that physician characteristics, such as specialty and age [24] and patient income [4] might have a significant effect on antibiotic prescribing during pregnancy. Evidence of this association is limited in mainland China. Further studies on the association between antibiotic prescriptions and prescriber and patient characteristics are needed.

We found that FDA category B was the most commonly prescribed antibiotic category during pregnancy in mainland China, whereas category D antibiotics with positive evidence of fetal risk only accounted for a very low proportion. Furthermore, our results showed a shift to the prescription of relatively safe antibiotics during pregnancy. In the first trimester category B antibiotics accounted for only 52.4% of all antibiotics and category D antibiotics accounted for 3.3% of all antibiotics; however, the prescription percentage of category B antibiotics increased to 97.6% and category D antibiotics accounted for only 0.2% of all antibiotics in the third trimester. Several studies have suggested that pregnant women in the first trimester are more prone to receiving drugs that are potentially harmful to the fetus [24,32,33,34] and this was likely due to the unawareness of pregnancy by the physicians and women in early pregnancy [24]. Category C antibiotics accounted 20% of all prescribed antibiotics, of which 50% was levofloxacin. Evidence of teratogenicity risk of levofloxacin is scarce. Available evidence for other fluoroquinolones suggests that the risk of major malformations is low [18]. However, they are generally contraindicated in pregnancy [1] and the current guidelines in China state that fluoroquinolones should be avoided during pregnancy [35]. It has been shown that fluoroquinolones may be associated with fetal risks such as renal toxicity, cardiac defects and central nervous system toxicity [1]. Furthermore, in 2016, the FDA added box warnings to all fluoroquinolones for the risk of disabling and harmful effects on the tendons, muscles, joints, nerves and central nervous system [36]. Therefore, fluoroquinolones are only recommended for those who do not have alternative treatment options during pregnancy [1,18]. Ornidazole, which had not been assigned to a FDA category, was the most commonly prescribed antibiotic for pregnant women in mainland China. This pattern was different from that reported in some developed countries, where specific kinds of antibiotics in category B were the most commonly used, such as nitrofurantoin, penicillins, azithromycin and metronidazole [3,11,27,37]. Evidence of effects of ornidazole on fetus from human studies is limited. Fetal risks of other nitroimidazoles are diverse, with metronidazole being classified as category B and tinidazole as category C due to limited evidence from human pregnancy studies [1,12]. Therefore, ornidazole should be used similar to tinidazole, which is recommended for use only if metronidazole fails to eradicate the infection [12,18].

### 4.3. Policy Implications

The appropriate use of antibiotics is a key strategy to balance the risk-versus-benefit decisions of drug use in pregnancy and to tackle antibiotic resistance. Available evidence and unknown information regarding the effects of antibiotics on pregnancy and fetus should be carefully considered when prescribing these drugs [7]. Although outpatient antibiotic prescriptions during pregnancy was not highly prevalent in mainland China, a substantial proportion of antibiotics were prescribed without adequate indications, while a large number of infectious conditions might have been undertreated. Diagnostic uncertainties and etiological diversity are major causes for inappropriate antibiotic use [16,38]; thus, point-of-care diagnostic tests that can rapidly distinguish different pathogens of common infections are useful for more accurate diagnosis and appropriate antibiotic selection [39]. In addition, more attention should be paid to high-risk populations, such as pregnant women aged >35 years and patients in emergency departments and secondary hospitals, when developing policies to control antibiotic prescriptions during pregnancy.

Although the majority of the prescribed antibiotics for pregnant women were FDA category B antibiotics, a considerable proportion of antibiotics with little evidence of fetal risk were prescribed during pregnancy in mainland China. Early pregnancy was the period with higher risk of excessive antibiotic prescriptions and unsafe antibiotic use. Physicians should pay attention to the marriage and pregnancy status when prescribing antibiotics to women of childbearing age. An alarm mechanism embedded in medical information systems to indicate fetal risk of drugs may be useful in reducing unsafe antibiotic prescriptions. Insufficient safety evidence may be an important factor leading to the use of drugs that are unsafe during pregnancy and underuse of safer drugs [37]. Hence, more studies are urgently needed to obtain evidence on the safety of drugs during pregnancy [32], especially for drugs that are commonly used and with unclear fetal risk, such as ornidazole. Furthermore, improving awareness and knowledge of fetal risks of antibiotics among physicians and pregnant women, combined with enhancing communication between doctors and patients, are potentially useful measures to reduce inappropriate and unsafe antibiotic use during pregnancy [32,40,41].

### 4.4. Strengths and Limitations

Our study has several strengths. We provided the most recent and comprehensive data on antibiotic use during pregnancy in ambulatory care settings in mainland China using a large and nationally representative sample of prescription data. Using a well-established and validated diagnosis classification scheme [15], our study provided an estimation of percentages of antibiotic prescriptions for co-occurring diagnoses of pregnancy, the proportion of potential inappropriate antibiotic prescriptions and the prescribing pattern of antibiotics belonging to different FDA pregnancy categories.

There were some limitations. First, our study was based on cross-sectional data and outpatient visits in different hospitals or on different dates of the same patient could not be linked together; thus, the percentage of antibiotic prescriptions might be underestimated and cannot be regarded as antibiotic use throughout pregnancy. Further studies based on a cohort design and a nationally representative population are needed. Second, some prescriptions that lacked standardization, such as incomplete diagnosis writing, might have been omitted or classified as inappropriate for antibiotic use, resulting in the omission of pregnancy visits and potential bias in the assessment of inappropriate antibiotic prescriptions. Third, as in our previous study [16], this study had no access to prescription data in primary care settings, private hospitals, or specialist hospitals. Antibiotic use during pregnancy in these medical institutions is still scarce. Fourth, in this study, we used only indications to assess the appropriateness of antibiotic prescriptions; other information such as administration route, dosage and therapy duration were not considered. Finally, the FDA categories are not designed to describe teratogenic risk, but are frequently misinterpreted as a graded system. Therefore, the complexity of drug use during pregnancy cannot be sufficiently addressed using such a simple system [8]. This system has been widely criticized for its oversimplification and incompetence to guide clinical practice and thus, it was replaced by a new narrative labelling system in 2015 [8,12]. Our results of antibiotic prescription patterns based on FDA pregnancy categories can only provide a rough description of fetal-risk considerations of antibiotic prescriptions during pregnancy. A more comprehensive evaluation scheme and further evidence of antibiotic safety during pregnancy are needed to further assess antibiotic use during pregnancy.

## 5. Conclusions

In mainland China, outpatient antibiotic prescriptions during pregnancy were not highly prevalent, with a percentage of 2.0%. Patient- and facility-level characteristics were associated with antibiotic prescriptions during pregnancy. Some conditions that do not justify antibiotic use were prescribed a considerable number of antibiotics and nearly three-quarters of antibiotics might have been prescribed without adequate indications to pregnant women. Antibiotics that fetal safety has not been sufficiently proven were widely used in pregnancy. Antibiotic stewardship efforts to optimize outpatient antibiotic prescriptions and reduce the use of potentially unsafe antibiotics are needed in China. Additional studies are needed to explore antibiotic prescriptions in primary care settings and determine more potential influencing factors, such as physicians’ characteristics, associated with antibiotic prescriptions during pregnancy.

## Figures and Tables

**Figure 1 antibiotics-10-00601-f001:**
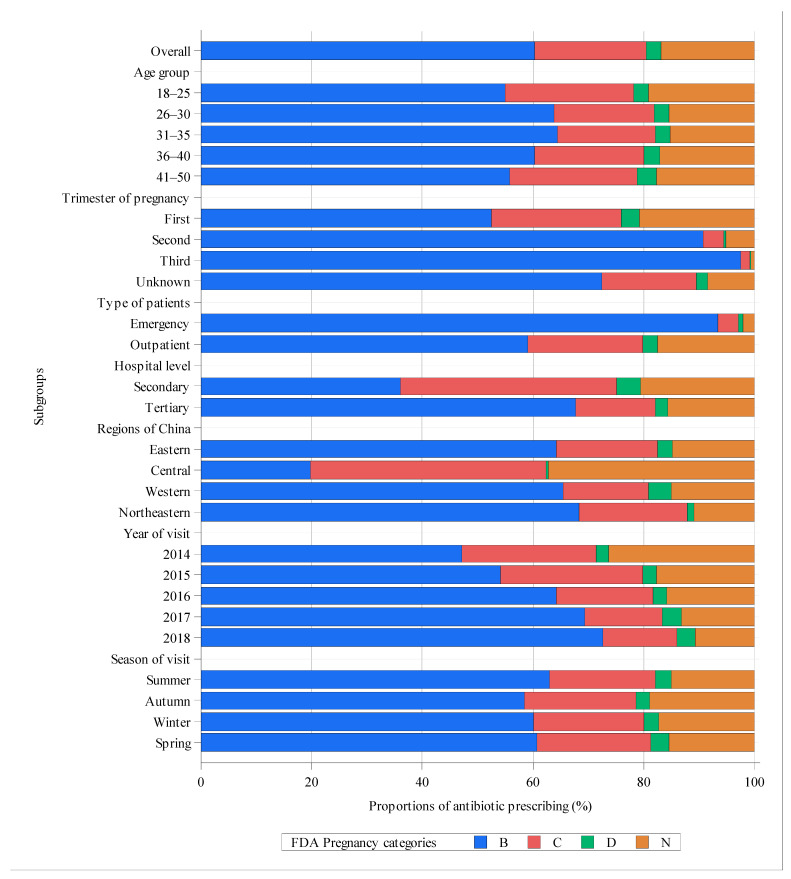
Proportions of antibiotics of FDA pregnancy categories within different subgroups. Category N is not a FDA pregnancy category. In this study, category N referred to antibiotics that had not been assigned any FDA category.

**Figure 2 antibiotics-10-00601-f002:**
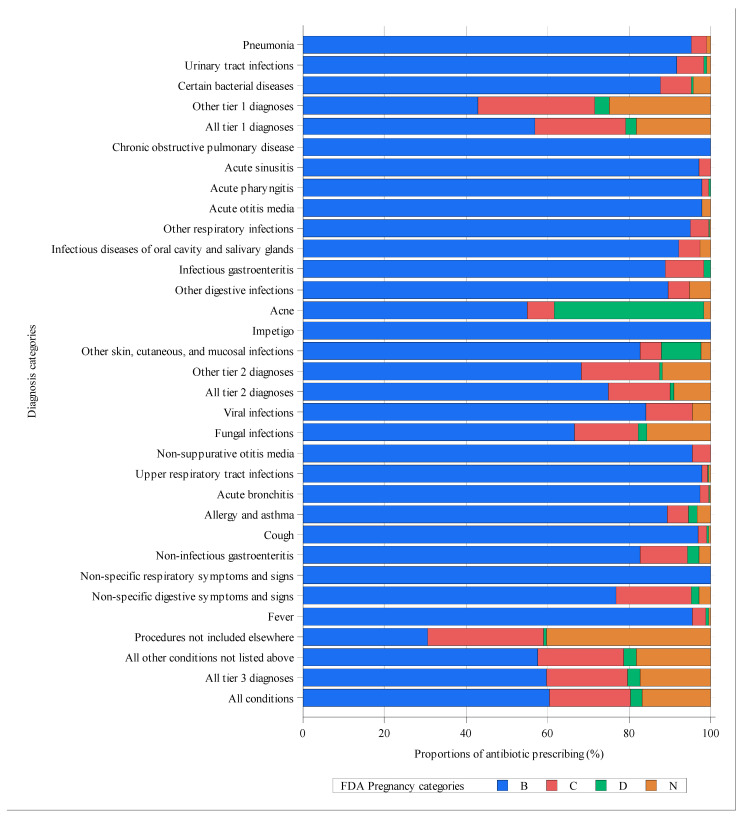
Proportions of antibiotics of FDA pregnancy categories within different diagnosis categories. Category N is not a FDA pregnancy category. In this study, category N referred to antibiotics that had not been assigned any FDA category.

**Table 1 antibiotics-10-00601-t001:** Basic characteristics of ambulatory care visits during pregnancy.

Subgroups	No. of Visits	Percentage, % *
Age group		
18–25	857,366	18.7
26–30	2,080,643	45.5
31–35	1,151,593	25.2
36–40	407,372	8.9
41–50	77,987	1.7
Trimester of pregnancy		
First	1,189,055	26.0
Second	870,274	19.0
Third	1,165,475	25.5
Unknown	1,350,157	29.5
Type of patients		
Emergency department	72,334	1.6
Outpatient department	4,502,627	98.4
Hospital level		
2nd	295,109	6.5
3rd	4,279,852	93.5
Regions of China		
Eastern	2,317,388	50.7
Central	201,805	4.4
Western	1,611,925	35.2
Northeastern	443,843	9.7
Year of visit ^a^		
2014	278,794	6.1
2015	1,127,874	24.7
2016	1,540,063	33.7
2017	1,278,681	27.9
2018	349,549	7.6
Season of visit		
Autumn	1,165,274	25.5
Winter	1,202,141	26.3
Spring	1,179,606	25.8
Summer	1,027,940	22.5
Tier of co-diagnoses ^b^		
Tier 1	35,442	0.8
Tier 2	109,165	2.4
Tier 3 or no co-diagnosis	4,430,354	96.8
Overall	4,574,961	100

* For rounding reasons, the sum of the percentages of some subgroups might not be exactly equal to 100%. ^a^ For years 2014 and 2018, only prescriptions in October, November and December of 2014 and January, February, March and April of 2018 were available. ^b^ Tier 1 diagnoses are conditions for which antibiotic is almost always indicated, such as pneumonia; tier 2 diagnoses are conditions for which antibiotic may be indicated, such as sinusitis; finally, tier 3 diagnose are all other conditions for which antibiotic is almost never indicated.

**Table 2 antibiotics-10-00601-t002:** Percentages of antibiotic prescriptions for various subgroups of ambulatory care visits by pregnant women.

Subgroups	Visits with Antibiotics Prescribed	Percentage of Antibiotic Prescriptions, % (95% CI)	OR (95% CI)
Overall	92,514	2.0 (2.0–2.0)	—
Age group			
18–25	27,380	3.2 (3.2–3.2)	0.38 (0.36–0.39)
26–30	30,345	1.5 (1.4–1.5)	0.25 (0.24–0.26)
31–35	20,216	1.8 (1.7–1.8)	0.33 (0.32–0.35)
36–40	10,787	2.6 (2.6–2.7)	0.52 (0.50–0.55)
41–50	3786	4.9 (4.7–5.0)	1
Trimester of pregnancy			
Second	9270	1.1 (1.0–1.1)	0.44 (0.43–0.45)
Third	5221	0.5 (0.4–0.5)	0.22 (0.21–0.23)
Unknown	14,129	1.1 (1.0–1.1)	0.30 (0.29–0.31)
First	63,894	5.4 (5.3–5.4)	1
Type of patients			
Emergency department	3742	5.2 (5.0–5.3)	8.68 (8.30–9.07)
Outpatient department	88,772	2.0 (2.0–2.0)	1
Hospital level			
3rd	76,113	1.8 (1.8–1.8)	0.38 (0.23–0.61)
2nd	16,401	5.6 (5.5–5.6)	1
Regions of China			
Eastern	38,780	1.7 (1.7–1.7)	0.19 (0.09–0.37)
Central	7826	3.9 (3.8–4.0)	0.43 (0.18–1.02)
Western	33,662	2.1 (2.1–2.1)	0.41 (0.20–0.85)
Northeastern	12,246	2.8 (2.7–2.8)	1
Year of visit ^a^			
2015	29,824	2.6 (2.6–2.7)	0.66 (0.64–0.68)
2016	25,975	1.7 (1.7–1.7)	0.58 (0.56–0.59)
2017	21,273	1.7 (1.6–1.7)	0.58 (0.57–0.60)
2018	6163	1.8 (1.7–1.8)	0.69 (0.66–0.72)
2014	9279	3.3 (3.3–3.4)	1
Season of visit			
Autumn	22,614	1.9 (1.9–2.0)	0.99 (0.97–1.02)
Winter	26,676	2.2 (2.2–2.2)	1.14 (1.12–1.17)
Spring	24,520	2.1 (2.1–2.1)	1.07 (1.05–1.09)
Summer	18,704	1.8 (1.8–1.8)	1
Tier of co-diagnoses ^b^			
Tier1	14,313	40.4 (39.9–40.9)	31.98 (30.95–33.05)
Tier2	8590	7.9 (7.7–8.0)	11.28 (10.97–11.60)
Tier3 or no co-diagnosis	69,611	1.6 (1.6–1.6)	1

^a^ For years 2014 and 2018, only prescriptions in October, November and December of 2014 and January, February, March and April of 2018 were available. ^b^ Tier 1 diagnoses are conditions for which antibiotic is almost always indicated, such as pneumonia; tier 2 diagnoses are conditions for which antibiotic may be indicated, such as sinusitis; finally, tier 3 diagnose are all other conditions for which antibiotic is almost never indicated.

**Table 3 antibiotics-10-00601-t003:** Percentages of antibiotic prescriptions for various diagnosis categories.

Diagnosis Categories *	No. of Visits	Visits with Antibiotics Prescribed	Percentage of Antibiotic Prescriptions in Pregnancy, % (95% CI)	Percentage of Antibiotic Prescriptions of the General Patients, % ^†^
Tier 1 diagnoses that antibiotics are almost always indicated				
Pneumonia	294	170	57.8 (52.0–63.5)	66.4
Urinary tract infections	5163	1635	31.7 (30.4–33.0)	51.1
Certain bacterial diseases	6959	3287	47.2 (46.1–48.4)	46.2
Other bacterial infections	23,026	9221	40.1 (39.4–40.7)	27.6
All tier 1 diagnoses	35,442	14313	40.4 (39.9–40.9)	42.2
Tier 2 diagnoses that antibiotics are sometimes indicated				
COPD	40	3	7.5 (1.6–20.4)	25.1
Acute sinusitis	262	99	37.8 (31.9–44.0)	37.8
Acute pharyngitis	2279	781	34.3 (32.3–36.3)	43.1
Acute otitis media	181	43	23.8 (17.8–30.6)	43.4
Other infectious diseases of the respiratory system	2328	770	33.1 (31.2–35.0)	35.7
Infectious diseases of oral cavity and salivary glands	1966	374	19.0 (17.3–20.8)	29.5
Infectious gastroenteritis	826	113	13.7 (11.4–16.2)	34.2
Other infectious diseases of the digestive system	510	48	9.4 (7.0–12.3)	9.0
Acne	505	55	10.9 (8.3–13.9)	29.3
Impetigo	13	2	15.4 (1.9–45.4)	30.3
Other skin, cutaneous and mucosal infections	1114	144	12.9 (11.0–15.0)	37.6
Other infectious diseases that antibiotic may be indicated	99,141	6158	6.2 (6.1–6.4)	26.4
All tier 2 diagnoses	109,165	8590	7.9 (7.7–8.0)	30.6
Tier 3 diagnoses that antibiotics are not indicated				
Viral infections	7638	33	0.4 (0.3–0.6)	2.7
Fungal infections	1654	35	2.1 (1.5–2.9)	3.4
Non-suppurative otitis media	91	21	23.1 (14.9–33.1)	36.2
Viral upper respiratory tract infection	8984	2808	31.3 (30.3–32.2)	40.9
Influenza	9	0	0.0 (0.0–33.6)	14.4
Acute bronchitis	2177	968	44.5 (42.4–46.6)	55.6
Allergy and asthma	1779	82	4.6 (3.7–5.7)	7.5
Cough	1240	179	14.4 (12.5–16.5)	23.6
Other non-infectious gastroenteritis	3139	304	9.7 (8.7–10.8)	12.2
Non-specific symptoms, signs of respiratory system	592	41	6.9 (5.0–9.3)	9.1
Non-specific symptoms, signs of digestive system	6209	247	4.0 (3.5–4.5)	14.4
Fever	1365	393	28.8 (26.4–31.3)	48.9
Procedures and surgeries not included elsewhere	8264	344	4.2 (3.7–4.6)	7.6
All other conditions not listed above	4,387,213	64,156	1.5 (1.5–1.5)	3.9
All tier 3 diagnoses	4,430,354	69,611	1.6 (1.6–1.6)	4.5
All conditions	4,574,961	92,514	2.0 (2.0–2.0)	10.9

* Tier 1 diagnoses are conditions for which antibiotic is almost always indicated, such as pneumonia; tier 2 diagnoses are conditions for which antibiotic may be indicated, such as sinusitis; finally, tier 3 diagnose are all other conditions for which antibiotic is almost never indicated. More details of diagnosis categories can be found in previous studies [15,16]. **^†^** These data were cited from a previous study [16]. General patients including pregnant women and all other patients of different visit reasons.

## Data Availability

Aggregate de-identified patient data with geographical region of outpatient visits masked to the level of China economic region divisions will be available after publication of this study. Requests should be sent to the corresponding authors and the management committee of the data center, who will discuss the requests and decide whether to share the data on the basis of the feasibility, novelty, and scientific rigor of the proposal. All applicants will need to sign a data access agreement.

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
