# Peer review of "Antibiotic Prescriptions among China Ambulatory Care Visits of Pregnant Women: A Nationwide Cross-Sectional Study"

_antibiotics, 2021, doi:10.3390/antibiotics10050601_

Round 1

Reviewer 1 Report

I have read your paper with great interest, and value the effort. However, I do have some additional suggestions and reflections

Prescription does not equals antibiotic consumption/use: is a prescription needed to buy antibiotics ? both words are currently used interchangeably, and that is not correct. I understood from other reports that there is additional access to antibiotics, or that there is already use of ‘remnants’.

How does the prescription practice during pregnancy differs from a non-pregnant cohort of young women ? higher, lower or similar ?

The FDA A,B,C, D and X classification is outdated, as the FDA shifted to a more descriptive part of this section of the SmPC. This should at least be explicitly mentioned?

Data source and participants: can you quantify the ‘population’ excluded in addition to the number of hospitals, and was this ‘structural’ so that specific regions are excluded, or ‘regional’ versus ‘cities’  ?

Discussion: how to understand the statement on unawareness of early pregnancy, as pregnancy was a registered diagnosis ? (line 436)

Author Response

Point 1: I have read your paper with great interest, and value the effort. However, I do have some additional suggestions and reflections

Prescription does not equals antibiotic consumption/use: is a prescription needed to buy antibiotics? both words are currently used interchangeably, and that is not correct. I understood from other reports that there is additional access to antibiotics, or that there is already use of ‘remnants’.

Response 1: Thanks for raising these points. In this study, the drug information was from the billing system, which means that all prescribed antibiotics were eventually dispensed to patients. In Chines hospitals drugs cannot be dispensed to patients without a prescription. However, non-prescription antibiotic purchasing can occur in retail pharmacies rather than hospitals.

We added more descriptions of the data link (page 3, line 88-91). We have further changed ‘antibiotic use’ to ‘antibiotic prescriptions’ when necessary to avoid misunderstanding.

Point 2: How does the prescription practice during pregnancy differs from a non-pregnant cohort of young women ? higher, lower or similar ?

Response 2: Thanks for raising these points. The antibiotic prescription rate for non-pregnant women aged between 18 and 45 was about 9.5%, much higher than that of pregnant women. However, describing the antibiotic prescriptions for non-pregnant women was beyond the scope of this study. Further, we have reported the antibiotic prescriptions in the general outpatients in our previous study published in the Lancet Infectious Diseases (doi:10.1016/S1473-3099(20)30596-X). In the Table 3, we gave the comparison between the pregnant women and general outpatients. In the Discussion (page 15-16, line 314-326), we discussed the differences between the pregnant women and general outpatients.

Point 3: The FDA A,B,C, D and X classification is outdated, as the FDA shifted to a more descriptive part of this section of the SmPC. This should at least be explicitly mentioned?

Response 3: We thank the reviewer for highlight this important issue. We have discussed the limitation of the FDA letter categorisation system and the newly application of the more narrative labelling system in the Discussion of Strengths and Limitations (page 18-19, line 412-420).

Yes, the FDA categories have been replaced by a new narrative labelling system in 2015. Because the previous letter classification system has been widely criticised for its oversimplification and incompetence to guide clinical practice. The FDA letter classification system was used in this study mainly for three reasons: (1) the new narrative labelling system has been rarely applied in the study of potential fetal risk of drugs during pregnancy; (2) it’s convenient to classify antibiotics into different categories by using the FDA A, B, C, D and X categories system; (3) the latest Chinese guidelines for clinic use of antibiotics (Reference 35) still use the FDA letter categories. Therefore, our results can be compared with other studies using the same classification system and can be easily understood by clinicians in China.

Point 4: Data source and participants: can you quantify the ‘population’ excluded in addition to the number of hospitals, and was this ‘structural’ so that specific regions are excluded, or ‘regional’ versus ‘cities’  ?

Response 4: Thanks for highlight this important issue. There were not structural differences between included and excluded patients. As a matter of fact, we have given more detailed descriptions of the data source in our previous studies published in the Lancet Infectious Diseases (doi:10.1016/S1473-3099(20)30596-X). In the Appendix 1 of the supplementary materials of that paper, we gave details of the recruitment process of the hospitals and the representativeness of the database. Further, in the Appendix 5, we provided comparisons between the included and excluded hospitals. Therefore, we cited that paper to describe the data source in this study (page 3, line 84-88; page 4, line 101) rather than repeated these descriptions in this paper.

Point 5: Discussion: how to understand the statement on unawareness of early pregnancy, as pregnancy was a registered diagnosis ? (line 436)

Response 5: We thank the reviewer for highlight this important issue. This study could not reach this conclusion. This statement was cited from the Reference 24. In order to avoid misunderstanding, we have changed this sentence to “Early pregnancy was the period with higher risk of excessive antibiotic prescriptions and unsafe antibiotic use” (page 17, line 382-383).

Reviewer 2 Report

This is an interesting paper to investigate antibiotic prescriptions during pregnancy in ambulatory care settings. We offer the following points for potential improvements.

Introduction

Page 2 line 66-70

1.The authors should describe the current status of research related to the use of antimicrobials among pregnant women worldwide.

2.The authors mentioned that only a few studies have investigated the use of antibiotics by pregnant women in China. What conclusions did these studies draw?

Materials and Methods

  1. Why were pregnant women aged 18 to 50 years selected for the study?
  2. It would be good to make subgroup analysis on the elderly pregnant women over 35 years old.

Results

Page 11 line 291-293

5. “… and the most commonly prescribed category B antibiotics were azithromycin (14.0% of all category A antibiotics), …”.  There is an error here.

Author Response

Point 1: This is an interesting paper to investigate antibiotic prescriptions during pregnancy in ambulatory care settings. We offer the following points for potential improvements.

Introduction

Page 2 line 66-70

1.The authors should describe the current status of research related to the use of antimicrobials among pregnant women worldwide.

Response 1: Thanks for rising this point. We have added a brief description of the current status of researches related to the antibiotic use during pregnancy worldwide (page 2, line 55-56; page 3, line 74-76) according to our literature review.

Point 2: 2.The authors mentioned that only a few studies have investigated the use of antibiotics by pregnant women in China. What conclusions did these studies draw?

Response 2: Thanks for rising this point. This statement was from our literature review of the studies on antibiotic use during pregnancy in China. We researched the PubMed for articles published to April 21, 2019 by using the terms ‘China’ (OR ‘Chinese’), ‘pregnancy’ (or ‘pregnant’), and ‘antibiotic’ (‘antibacterial’, or ‘antimicrobial’). A total of 129 articles were founded, but none of these studies focused on the antibiotic prescriptions during pregnancy. Further, a Chinese literature database CNKI were searched by using a similar strategy. A total of 137 articles were founded, but only two studies investigated the antibiotic prescriptions during pregnancy in a single hospital.

To avoid over-interpretation, we have deleted this statement and kept the other one -- “Evidence of the use and appropriateness of antibiotics during pregnancy is particularly scarce at the national level in China” (page 3, line 75-76). Because to the best of our knowledge, this study is the first to investigate the antibiotic prescriptions during pregnancy at the national level in China.

Point 3: Materials and Methods

Why were pregnant women aged 18 to 50 years selected for the study?

It would be good to make subgroup analysis on the elderly pregnant women over 35 years old.

Response 3: Thanks for rising this point. According to the China Population and Employment Statistical Yearbook, childbearing age is between 15 and 49. Teenage pregnancies are very rare and are likely to be terminated later. Thus we finally included pregnant women aged 18-50.

We have added subgroup analyses of age in the supplementary materials Table A2.

Antibiotic prescription rates for different age groups and diagnosis categories were reported in the Table A2 and in the main text (page 10, line 228-233). Further, in the Figure 1, there was already subgroup analyses for the proportions of FDA categories within each age group.

Point 4: Results

Page 11 line 291-293

“… and the most commonly prescribed category B antibiotics were azithromycin (14.0% of all category A antibiotics), …”. There is an error here.

Response 4: Thanks for rising this point. We have corrected this error (page 12, line 265-266).
